# A Method for Rapid Screening, Expression, and Purification of Antimicrobial Peptides

**DOI:** 10.3390/microorganisms9091858

**Published:** 2021-09-01

**Authors:** Yingli Zhang, Zhongchen Li, Li Li, Ben Rao, Lixin Ma, Yaping Wang

**Affiliations:** 1State Key Laboratory of Biocatalysis and Enzyme, Engineering Hubei Collaborative Innovation Center for Green Transformation of Bio-Resources, Hubei Key Laboratory of Industrial Biotechnology, Biology Faculty of Hubei University, Hubei University, Wuhan 430062, China; zyl19980703@163.com (Y.Z.); lizc0408@163.com (Z.L.); groundcontrol2021@163.com (L.L.); malixing@hubu.edu.cn (L.M.); 2National Biopesticide Engineering Technology Research Center, Hubei Biopesticide Engineering Research Center, Hubei Academy of Agricultural Sciences, Biopesticide Branch of Hubei Innovation Centre of Agricultural Science and Technology, Wuhan 430064, China; raoben1983729@aliyun.com

**Keywords:** antimicrobial peptides expression, CL7 tag, *Escherichia coli*, *Pichia pastoris*

## Abstract

In this study, a method for the rapid screening, expression and purification of antimicrobial peptides (AMPs) was developed. AMP genes were fused to a heat-resistant CL7 tag using the SLOPE method, and cloned into *Escherichia coli* and *Pichia pastoris* expression vectors. Twenty *E. coli* and ten *P. pastoris* expression vectors were constructed. Expression supernatants were heated, heteroproteins were removed, and fusion proteins were purified by nickel affinity (Ni-NTA) chromatography. Fusion proteins were digested on the column using human rhinovirus (HRV) 3C protease, and AMPs were released and further purified. Five AMPs (1, 2, 6, 13, 16) were purified using the *E. coli* expression system, and one AMP (13) was purified using the *P. pastoris* expression system. Inhibition zone and minimum inhibitory concentration (MIC) tests confirmed that one *P. pastoris*¬-derived and two *E. coli*-derived AMPs have the inhibition activity. The MIC of AMP 13 and 16 from *E. coli* was 24.2 μM, and the MIC of AMP 13 from *P. pastoris* was 8.1 μM. The combination of prokaryotic and eukaryotic expression systems expands the universality of the developed method, facilitating screening of a large number of biologically active AMPs, establishing an AMP library, and producing AMPs by industrialised biological methods.

## 1. Introduction

Antimicrobial peptides (AMPs) are small, naturally occurring peptides 5–100 amino acids in length, and possess high structural diversity [1,2,3]. Structures include α-helices and β-sheet regions rich in cysteine, and regions containing a high percentage of specific amino acids with unique structures. AMPs are produced in various eukaryotic and prokaryotic organisms, and some affect the innate immune systems of bacteria, plants, insects and mammals.

Recently, there has been a surge in the use of small peptides as drug candidates because they provide significant advantages over traditional small molecule drugs. AMPs have the advantages of low molecular weight, high heat resistance, broad-spectrum antimicrobial activity and, importantly, they show minimal antimicrobial resistance, making them a promising treatment for antimicrobial therapy [4].

AMPs protect against a wide variety of pathogens, including prokaryotic microorganisms, fungi, viruses, protozoa, insect cells and cancer cells. Advantages include antibacterial activity, degradability in vivo and, because they are from natural sources, they are generally safe and have non-toxic effects and, hence, are applicable in foods and feed additives. However, there are also some limitations for AMP applications. The natural AMPs are labile, depending on the surrounding environments, such as the presence of protease, pH change, and so on. The potential toxicity of AMPs for oral application and the high cost of peptide production are the other obstacles [5,6].

There are various ways to obtain AMPs [7,8,9,10,11,12]: (1) They can be obtained directly from animals and plants, but this has the disadvantages of high cost, low yield, cumbersome technical operations and often insufficient purity. (2) They can be prepared by chemical synthesis, such as solid-phase synthesis, fragment condensation and other synthetic peptide procedures. Chemically synthesised AMPs are generally of good purity and are suitable for the production of high-value pharmacological-grade peptides, but they have disadvantages including expensive synthesis and low chemical reaction efficiency. (3) Enzymatic hydrolysis using proteases can yield AMPs with higher activity than other methods. However, since the target sequence or structure is not artificially designed, purification may be difficult and costly. (4) Finally, genetic engineering technology can be used to produce AMPs, and this may eventually become the main method.

Both eukaryotic and prokaryotic systems are currently used for heterologous expression of AMPs [13]. The characteristics of various AMPs, including structure, disulfide bond abundance, acidic/basic residue distribution, isoelectric point and degree toxicity, all place different demands on the requirements for host selection. The most commonly used expression engineering strains are *Escherichia coli*, *Bacillus subtilis* and *Pichia pastoris*. In the past few decades, researchers have used several expression systems to produce AMPs in a cost-effective manner, and *E. coli* alone has been used to produce more than 80% of all recombinant AMPs. However, *E. coli* cells are not suitable for small peptides expressed at high concentrations, and the toxicity of AMPs to the host cannot be ignored.

The *P. pastoris* yeast strain has been developed into an excellent engineered host for high-concentration and high-purity heterologous expression of proteins, including short peptides from different sources [14,15]. This host has been successfully used to produce a variety of functional recombinant proteins from humans, animals, plants, fungi, bacteria and viruses. Most importantly, this host can express recombinant AMPs in secreted form, which provides several advantages over prokaryotic expression systems, including correct folding in the cell, formation of disulfide bonds, correct protein tertiary structure and other post-translational modifications that affect protein function. The secretion of heterologous proteins prevents their accumulation from affecting the normal growth of the host, and avoids intracellular protein contamination. In addition, extracellular secretion simplifies the purification steps, and high-density fermentation enables industrial production of AMPs. Due to these advantages of the *P. pastoris* expression system, it has become important in the industrial production of AMPs.

In this study, we developed a method for rapid screening, expression and purification of AMPs. We synthesised small peptides using the SLOPE method, which greatly improves the efficiency of constructing AMP expression vectors, and has lower costs. AMP genes were fused to a heat-resistant CL7 tag [16,17] using the SLOPE method, and cloned into *Escherichia coli* and *Pichia pastoris* expression vectors. Twenty *E. coli* and ten *P. pastoris* expression recombinants were constructed. The CL7 tag promotes solubilisation, good thermal stability and provides heat resistance to the fusion protein. Expression supernatants were heated, heteroproteins were removed, and fusion proteins were purified by nickel affinity (Ni-NTA) chromatography. Fusion proteins were digested on the column using high-efficiency low-cost human rhinovirus (HRV) 3C protease, and AMPs were released and further purified. Five AMPs (1, 2, 6, 13, 16) were screened and purified using the *E. coli* expression system, and one AMP (13) was purified using the *P. pastoris* expression system. Inhibition zone and minimum inhibitory concentration (MIC) tests confirmed that one *P. pastoris*-derived and two *E. coli*-derived AMPs have the inhibition activity. The MIC of AMP 13 and 16 from *E. coli* was 24.2 μM, and the MIC of AMP 13 from *P. pastoris* was 8.1 μM. The combination of prokaryotic and eukaryotic expression systems expands the universality of the developed method, facilitating screening of a large number of biologically active AMPs, establishing an AMP library and producing AMPs by industrialised biological methods.

## 2. Materials and Methods

### 2.1. Strains, Reagents and Media

*Pichia pastoris* GS115 and *E. coli* DH5α/BL21 strains were purchased from Invitrogen (Shanghai, China). The pHBM905BDM plasmid was constructed in our laboratory. All culture media, including Luria–Bertani (LB), minimal dextrose (MD), buffered minimal glycerol (BMGY) and buffered minimal methanol (BMMY) were prepared as described in the *P. pastoris* expression manual (Invitrogen).

### 2.2. Construction of the E. coli Expression Vector pET23a-HH3C

The starting vector was pET23a. A fusion gene containing a 6× HIS sequence (18 bp), a CL7 sequence (390 bp), and a human rhinovirus (HRV) 3C sequence (24 bp) was synthesised by Sangon BioTech (Shanghai, China). This fusion gene was inserted into the pET23a vector according to the method described by Li et al. [18], resulting in plasmid pET23a--HH3C. Using the same method, the fusion gene was inserted into the pPicZα and pHBM905BDM vectors, resulting in pPICZα-HH3C and pHBM905BDM-HH3C, respectively.

### 2.3. Synthesis of AMP Nucleotide Sequences

We selected AMP amino acid sequences from the AMP library website (http://aps.unmc.edu/AP/main.php (accessed on 16 October 2018)). The nucleotide sequences of these genes were optimised according to the codon preferences of *E. coli* and *P. pastoris*. We designed several pairs (2, 3, 4) of primers according to the AMP nucleotide sequence. These primers were then mixed, and a SLOPE reaction was performed as follows: 98 °C for 3min, 37 °C for 30min, 12 °C for 10min. After SLOPE reactions were performed, the products were used as templates. The universal vector pET23a-HH3C was amplified by inverse PCR to obtain the linearising vector [19,20]. Then, T4 DNA polymerase was utilized to treat the vector to obtain vector fragments with sticky ends. The AMP primers were equipped with adapters that are complementary to the sticky end of the vector. Importantly, each primer only needs to go through a single SLOPE reaction to obtain the product of the gene fragment to be used. The vector fragment treated with T4 polymerase and the SLOPE reaction product were mixed at a molar ratio of 1:3, and the ligated products were used to transform *E. coli* cells. The resulting plasmids were confirmed by PCR and sequencing.

### 2.4. Expression of AMPs in E. coli

The plasmids pET23a-HH3C-AMP were transformed into *E. coli* BL21. The recombinant strains were induced by IPTG. After induction, bacteria were collected in a 50 mM centrifuge tube, centrifuged at 9000 rpm for 10 min, and the supernatant was discarded. After adding 30 mM HRV 3C Cleavage Buffer (50 mM Tris, 150 mM NaCl, pH7.5) to resuspend cells, they were centrifuged, and this step was repeated once or twice. After washing, the cell pellet from 100 mL bacterial solution was resuspended in 20 mM HRV 3C Cleavage Buffer to prepare for ultrasonic cell disruption. Bacteria were lysed until the solution was clear, and samples were centrifuged at high speed (≥10,000 rpm) for 10 min. After centrifugation, the supernatant was subjected to SDS-PAGE to assess protein expression. In addition, as a control, the same volume (1 mL) of each bacterial supernatant was heated to 80 °C for 30 min. After high temperature treatment, the supernatant was centrifuged at high speed and the supernatant was collected for SDS-PAGE.

### 2.5. HRV 3C Cleavage

The nickel column was activated by HRV 3C Cleavage Buffer containing 10 mM imidazole, and then the protein sample was subjected to the column, which was washed three times with HRV 3C Cleavage Buffer, with 3 bed volumes each time. HRV 3C Cleavage Buffer was added to resuspend the nickel beads at a bed volume:reaction buffer (2ml HRV 3C Cleavage Buffer, 100 μL 100mM EDTA, 5 μL HRV 3C protease) volume ratio of 1:1. Digestion was performed at 4 °C on a slow shaker for 3 to 4 h. After digestion, the column was eluted with one bed volume of 1 M NaCl, and the flow-through was collected which was subjected to the following experiments. Then, the AMP concentrations were measured by the Bradford method.

### 2.6. Inhibition Testing

To measure AMP activity, the agarose plate method was employed. Microorganisms for inhibition testing were activated, 3–5 single colonies were picked and placed in liquid LB medium, cultured on a shaker at 37 °C for 6 h. Solid LB medium was melted by heating, 500 μL of liquid bacterial culture was added for every 100 mL of medium after cooling slightly, and the mixture was poured into a bacterial plate to solidify. Oxford cups were inserted into the solidified mixed bacterial plate, while being careful not to penetrate the medium completely, so as not to affect the penetration of the test liquid on the plate. Attention was paid to the spacing between the Oxford cups. Corresponding Oxford cups were marked and positive/negative controls (100 μL each) were added. Plates were incubated at 37 °C overnight to observe the antibacterial effects.

### 2.7. MIC Testing

When measuring the minimum inhibitory concentration (MIC) of the expressed and purified AMPs, we performed the test according to the Clinical and Laboratory Standards Association. Specifically, microorganisms to be tested were activated, and five colonies were picked and separately cultured into liquid medium on a shaker (220 rpm) at 37 °C until the absorbance at 600 nm (OD_600_) reached 0.13–0.17; thus, corresponding to a biomass concentration of ~1 × 108 colony-forming units (CFU)/mL. Samples for testing were diluted to 4 mM, and two-fold serial dilutions were performed in a 96-well plate, with the concentration ranging from 100 to 0.195 μM. Each dilution (50 μL) was performed in duplicate. Next, 50 μL of microbial culture containing ~1 × 10^6^ CFU/mL was added to each well. After incubating overnight at 37 °C, OD_600_ was measured using a microplate spectrophotometer. The MIC is defined as the minimum test compound concentration (μM) required to visibly inhibit the growth of microorganisms.

## 3. Results

### 3.1. Synthesis of AMP DNA Sequences Using SLOPE Technology

We used SLOPE technology to synthesise the nucleotide sequences of AMPs. SLOPE is used for the annealing of multiple primers to synthesise small fragments. This method avoids the disadvantages of expensive gene synthesis and the low efficiency of PCR amplification. It is suitable for small fragment molecular cloning.

We chose twenty AMPs (Table 1) from the antimicrobial peptide library website for production using the *E. coli* expression system. The DNA sequences of these AMPs were synthesised according to codon preference. The PCR results showed that the size of the AMP fragments obtained by the SLOPE reaction were consistent with the theoretical sizes of the corresponding AMPs (Figure 1).

These PCR products were purified and cloned into the pET23a-HH3C vector, resulting in pET23a-HH3C-AP(1-N). The recombinant plasmids were confirmed and transformed into *E. coli* BL21 cells, resulting in EAP(1-N) strains (Figure 2).

### 3.2. Expression of AMPs in E. coli and Purification

These recombinant EAP(1-N) strains were cultured in LB and induced with 0.5 mM IPTG at 16 °C. Expression of AMPs was assessed by SDS-PAGE (Figure 3). The results showed that AMPs (1, 2, 6, 13 and 16) were expressed at high levels in *E. coli* BL21 (DE3). These supernatants were incubated in a water bath at 80 °C for 30 min, centrifuged at high speed, filtered through a 0.45 μm filter, and subjected to Ni-NTA affinity chromatography. It showed that, after the high temperature treatment, heteroproteins in the supernatant were effectively reduced while target proteins were unaffected. The results showed that AMP13 and AMP16 saturated the column (Figure 4), and the amount of visible heteroproteins was low; hence, these target proteins were deemed suitable for subsequent studies. The fourth lane in Figure 4 shows that the target fusion protein bound successfully to the nickel column.

After fusion proteins were purified by Ni-NTA affinity chromatography, they were cleaved by HRV 3C to remove the CL7 protein. The collected components were analysed by Tricine-SDS-PAGE (Figure 5), and the results of restriction digestion were observed. Lanes 3, 4 and 5 show the digestion supernatant, HRV 3C Cleavage buffer and 1M NaCl, respectively. The HRV 3C protease band can be observed along with the target AMPs. The results showed that the size of the fusion protein was altered significantly before and after digestion. The larger bands in the digestion supernatant and the eluate after digestion correspond to the HRV 3C protein which is ~22 kDa in size. The band marked by the box in Figure 5 is the target AMP band. After digestion, only a small part of the target AMPs was present in the digestion supernatant.

Theoretically, target AMPs should be in the digestion supernatant. However, it was found that part of the AMPs could not be completely released into the digestion supernatant after digestion. It remained bound to the nickel column. Hence, 1 M NaCl was used to release the target AMPs as much as possible. The collected protein sample was incubated in a water bath at 80 °C for 30 min to remove heteroproteins and other contaminants introduced with the HRV 3C protease. Supernatants were collected after temperature treatment, concentrated, and desalted to obtain target AMPs. The protein concentration measured by the Bradford method was 0.2 mg/mL, 0.3 mg/mL, 0.2 mg/mL, 0.6 mg/ML and 0.7 mg/mL for AMP1, AMP 2, AMP 6, AMP 13 and AMP 16, respectively.

Judging from the purification results (Figure 6), the purity of the obtained proteins was good, and there were no other heteroproteins. The bands corresponding to the five target proteins were all larger than the theoretical value, consistent with the results of Tricine SDS-PAGE, and suggesting that target proteins may form multimers.

### 3.3. AMP Activity Testing

There were no inhibition zones around negative controls 1 and 2, but there was an inhibition zone around positive control 3 (Figure 7). There were no inhibition zones around test samples AMP1, AMP 2 and AMP 6, but there was an inhibition zone around samples AMP 13 and 16.

The MIC results (Figure 8) showed that the minimum increase in OD_600_ value for AMP1 was 128 mg/L, and the minimum increase in OD_600_ for AMP2 and AMP6 was >128 mg/L. This means that AMP1, AMP2 and AMP6 had no inhibition activity against testing the microorganisms. The minimum concentration of purified AMP13 and AMP16 that inhibited an increase in the OD_600_ of the bacterial solution was 32 mg/L and 64 mg/L, respectively. The MIC of AMP16 was 24.2 μM.

### 3.4. Expression of AMPs in P. pastoris and Purification

The *P. pastoris* expression plasmids pHBM905BDM-HH3C and pPICZα-HH3C were constructed similarly to the *E. coli* expression vector pET23a-HH3C. Fragments of AMP1, AMP2, AMP6, AMP13 and AMP16 were cloned into pHBM905BDM-HH3C and pPICZα-HH3C, resulting in ten recombinant plasmids. They were transformed into *P. pastoris*, resulting in corresponding recombinant strains PBEAP(N) and PZEAP(N), respectively. These strains were cultured and induced. The results of shake flask experiments showed that the expression supernatants of PBEAP (1, 2, 6, 13, 16) did not produce the expected AMPs. However, the results of PZEAP (1, 2, 6, 13, 16) experiments revealed a band of the expected product from PZEAP13 (Figure 9).

The supernatant of strain PZEAP13 was subsequently used for purification, and the results were assessed by SDS-PAGE (Figure 10) and Tricine SDS-PAGE (Figure 11). Comparative analysis before and after temperature treatment using SDS-PAGE (Figure 10, lanes 1 and 2) showed that high temperature treatment did not denature or decrease the amount of the target fusion protein; the change in the size of the protein in lanes 3 and 4 clearly show that AMPs were successfully released by HRV 3C protease. The specific cleavage site was cut (lanes 5 and 6), and there was a band corresponding to HRV 3C protease.

Similarly, Tricine-SDS-PAGE analysis showed that high temperature treatment did not denature or decrease the amount of the target fusion protein (Figure 11, lanes 1 and 2). The change in the size of the protein in lanes 3 and 4 clearly shows that AMPs were successfully recovered by HRV 3C protease. The specific cleavage site was cut, and bands marked by boxes in lanes 5 to 9 correspond to the target AMPs released following cleavage. Figure 12 also confirmed these results.

The *E. coli* DH5α mixed bacteria plate was prepared as described above. Two negative controls (HRV 3C Cleavage Buffer and 1 M NaCl) and a positive control (2 mg/mL ampicillin) were included. According to the growth of the mixed bacteria plate after culture, there were no inhibition zones around negative controls 1 and 2, but there was an inhibition zone around positive control 3. Apparently, AMP13 showed inhibition activity. Using the MIC measurement method described above, the minimum concentration of purified AMP13 produced by PZEAP13 that inhibited an increase in the OD_600_ of the bacterial solution was 16 mg/L, equating to an MIC for *E. coli* of 8.1 μM.

Figure 13B is a schematic diagram of the 96-well plate MIC experiment (data are from three parallel replicates). The horizontal direction is AMP13 from *P. pastoris*, the vertical direction is the final peptide concentration (128 mg/L to 0.25 mg/L), 11 is the negative control and 12 is the positive control (20 mg/L ampicillin). The results showed that the minimum concentration of AMP13 from *P. pastoris* that inhibited an increase in the OD_600_ of the bacterial solution was 16 mg/L.

## 4. Discussion

The AMP sequences chosen in this experiment were selected from an AMP library, cloned, expressed, and screened using a workflow that is universal. AMP genes were cloned into pET-23a, pPICZα and pHBM905BDM vectors, and expressed and screened using *E. coli* and *P. pastoris* systems. The purification process included high-temperature treatment of the protein supernatant, HRV 3C protease enzyme digestion on the nickel column, subsequent temperature treatment, protein concentration and desalination. This approach has multiple advantages: (1) The SLOPE method is superior to other methods. When obtaining AMP gene fragments using the SLOPE method, the high cost is greatly reduced compared with traditional gene synthesis, and it can generate gene fragments with sticky ends in one step. Following processing of the ready-to-use vector, the cloning efficiency can also be improved because the large-scale screening of AMPs can be achieved without PCR to obtain each target gene fragment. (2) Most of the AMPs and the CL7 fusion protein tag are resistant to high temperatures, which makes the purification process easier when using an *E. coli* expression system. After appropriate temperature treatment of the expression supernatant, most of the endogenous proteins are denatured and precipitated due to temperature intolerance, while the fusion protein can tolerate high temperature treatment. After high-speed centrifugation, some heteroproteins are precipitated and can be separated. This affords a useful preliminary purification step, which makes the procedure cheap, highly efficient and convenient. (3) The method is compatible with both *E. coli* and *P. pastoris* expression systems. The prokaryotic *E. coli* expression system has advantages, including a rapid growth cycle, low pollution and simple culture conditions, making it suitable for fast screening and expression of AMPs. The eukaryotic *P. pastoris* expression system is more suited to the secretion and expression heterologous proteins, especially where the copy number of the target gene sequence is increased, expression conditions are optimised and high-density fermentation is employed. These advantages make this system well suited to the industrial production of AMPs.

After construction of the *E. coli* expression vector, it was transformed into the *E. coli* BL21 (DE3) strain for expression in shake flasks. During this process, the number of colonies on each plate was not significantly different, while the morphology of colonies on plates containing samples 1, 2, 6, 7, 11, 13 and 16 did differ, and colonies were significantly smaller than for other samples. Bacterial culture took 8 h for the OD_600_ to reach 0.6 for EAP1, 2, 6, 11 and 16, and 9 h for EAP7 and 13. After induction with 1 mM IPTG for 24 h at 18 °C, the number of bacteria collected was significantly less for these samples than for others. Thus, we speculated that, although the fusion expression strategy was successfully employed, the fusion protein expressed by the host had a certain inhibitory effect on host cell growth. This phenomenon can be utilized to screen recombinant strains. If the amount of transformed plasmid and the volume of transformed competent cells are the same, then under the same culture conditions, smaller colonies on the transformed plate could express AMPs at higher levels than those larger colonies. With sufficient data, this feature could be used to initially select strains for expression, or to select colonies with better expression potential on the same transformation plate. Similarly, when preparing bacterial biomass in shake flasks, and all other conditions are optimised, shake flasks displaying a slower increase in the OD_600_ value are more likely to express larger amounts of heterologous fusion proteins. After the screening of highly expressed AMPs using *E. coli*, these AMP genes were cloned into the *P. pastoris* expression vectors pPICZα and pHBM905BDM, resulting in five constructs. The following experiments showed that only pPICZα- HH3C -13 successfully expressed the target protein.

Activity tests indicated that two AMPs produced by *E. coli* could yield inhibition zones, along with one AMP produced in the yeast expression system. In the MIC experiment, samples not yielding inhibition zones displayed no activity, consistent with the published literature. Importantly, the inhibition zone experiment requires high antibacterial activity. The AMPs with effective MICs using the *E. coli* expression system were EAP13 and EAP16. After purification of AMP13 from *P. pastoris*, we observed a clear circle of inhibition in the antibacterial experiment, and the effective MIC was 16 mg/mL. AMP1, AMP2 and AMP6 obtained by expression and purification in *E. coli* did not exhibit detectable antibacterial activity. We eliminated the possible influence of the expression conditions and purification process on antibacterial peptidase activity. After HRV 3C protease digestion, there were two amino acid residues (G and P) appended at the N-terminus of the AMPs that may affect its activity. The MICs of the purified AMPs obtained in this study were lower than those reported previously. AMP activity can be optimised by manipulating the expression conditions and the purification protocol. The purity of the obtained AMPs can greatly improve the MIC. The AMPs can also be characterized in depth, such as the effects of pH, temperature and metal ions on enzyme activity, along with the activity of AMPs against various Gram-negative and Gram-positive bacteria and fungi.

## Figures and Tables

**Figure 1 microorganisms-09-01858-f001:**
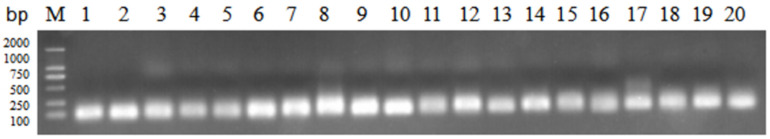
Agarose gel (1.5%) analysis of SLOPE products of AMPs. M, DL2000 DNA marker. Lane 1–20, represents AMPs 1–20 SLOPE products.

**Figure 2 microorganisms-09-01858-f002:**
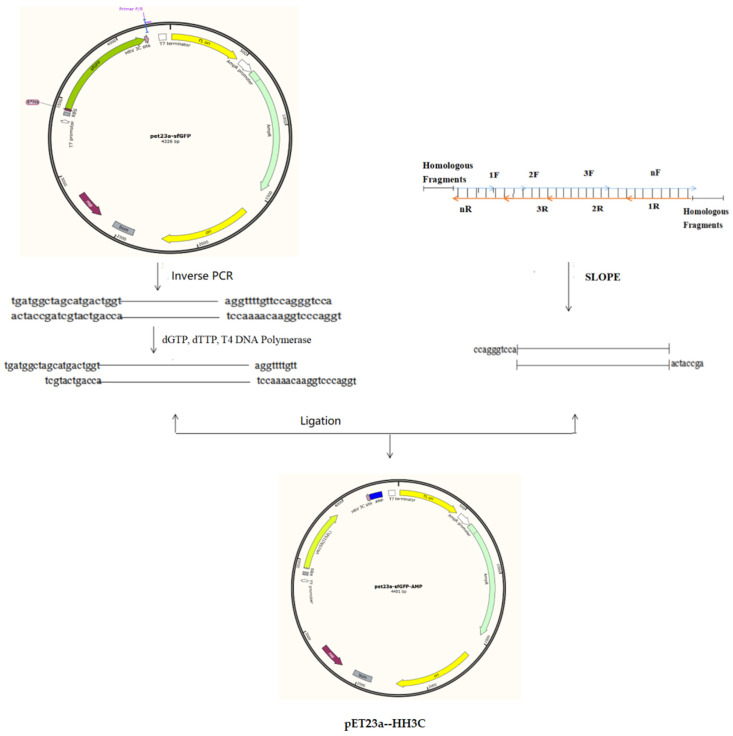
The construction of the AMP expression plasmids.

**Figure 3 microorganisms-09-01858-f003:**
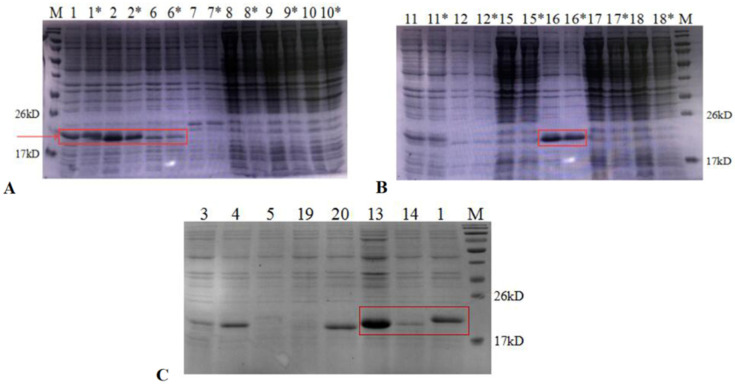
SDS-PAGE (12%) analysis of expression of AMPs in *E. coli*. (**A**,**B**), lane 1–18, the supernatants of EAP (1, 2, 6, 7, 8, 9, 10, 11, 12, 15, 16, 17, 18); lane 1–18*, the supernatant of EAP (1, 2, 6, 7, 8, 9, 10, 11, 12, 15, 16, 17, 18) heated to 80 °C for 30 min; (**C**), lane 1, 3, 4, 5, 13, 14, 19, 20, the supernatant of EAP (1, 3, 4, 5, 13, 14, 19, 20) heated to 80 °C for 30 min.

**Figure 4 microorganisms-09-01858-f004:**
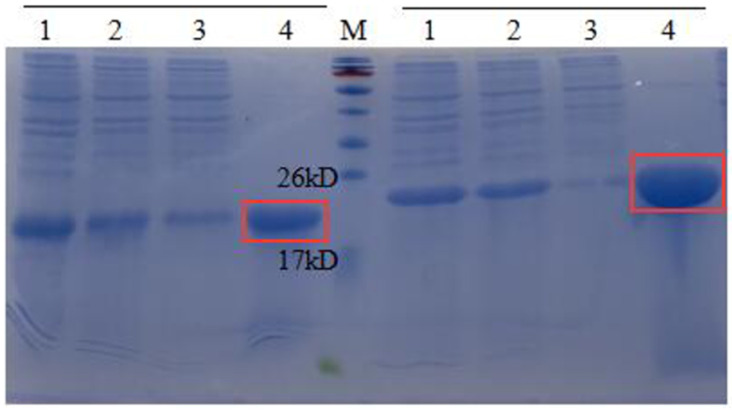
SDS-PAGE (12%) analysis of AMP (13, 16) purification. M, protein molecular weight marker. Left, lane 1, the supernatant of EAP 13; lane 2, the supernatant of EAP 13 heated to 80 °C for 30 min; lane 3, the flow-through of EAP 13 with Ni2+-affinity; lane 4, the AMP 13 purified with Ni2+-affinity. Right, lane 1, the supernatant of EAP 16; lane 2, the supernatant of EAP 16 heated to 80 °C for 30 min; lane 3, the flow-through of EAP 16 with Ni2+-affinity; lane 4, the AMP 16 purified with Ni2+-affinity.

**Figure 5 microorganisms-09-01858-f005:**
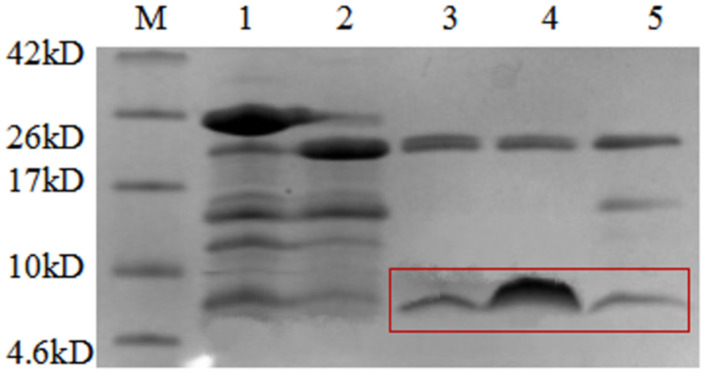
Tricine SDS-PAGE analysis of AMPs cleaved by HRV 3C. M, protein molecular weight marker. Lane 1, the nickel column before digestion; lane 2, the nickel column after digestion and elution; lane 3, the digestion supernatant; lane 4, HRV 3C Cleavage buffer; lane 5, 1M NaCl.

**Figure 6 microorganisms-09-01858-f006:**
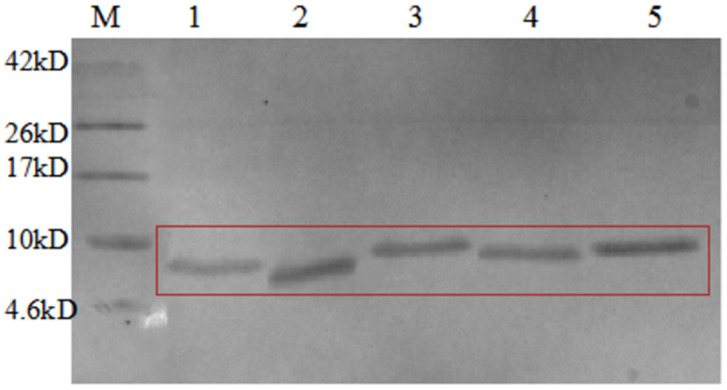
Tricine SDS-PAGE analysis of AMP purification. Lane 1-5, the purified AMP 1, 2, 6, 13 and 16.

**Figure 7 microorganisms-09-01858-f007:**
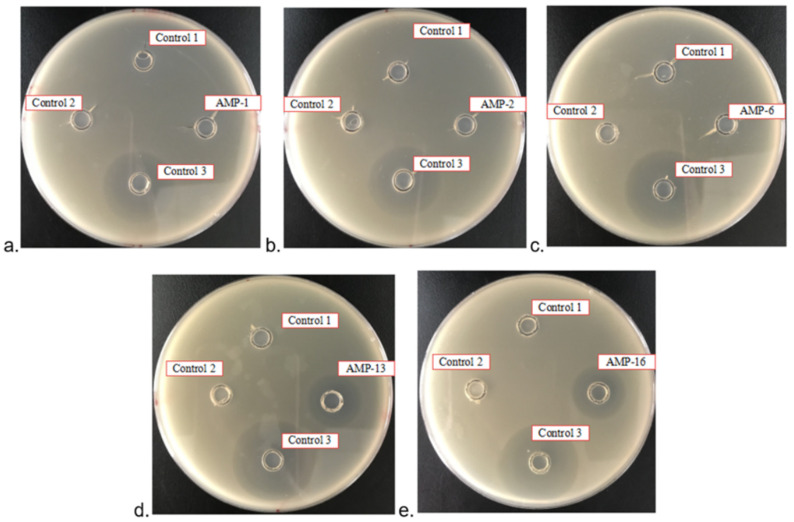
Inhibition zone tests of AMP 1, 2, 6, 13 and 16. Control 1, HRV 3C Cleavage Buffer; Control 2, HRV 3C Cleavage Buffer with 1 M NaCl; Control 3, Positive control, ampicillin antibiotic at a concentration of 200 mg/L. (**a**) Inhibition zone tests of AMP1; (**b**) Inhibition zone tests of AMP2; (**c**) Inhibition zone tests of AMP6; (**d**) Inhibition zone tests of AMP13; (**e**) Inhibition zone tests of AMP16. Every oxford cup contains 100 μL liquid.

**Figure 8 microorganisms-09-01858-f008:**
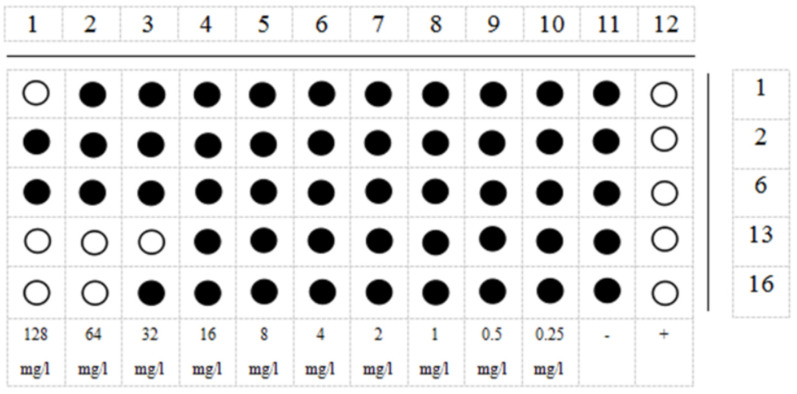
The schematic diagram of a 96-well plate measuring MIC (data taken from three parallel experiments). Horizontal direction is AMP 1, 2, 6, 13, 16 test rows; Vertical direction 1–10 is the final concentration of 128 mg/L to 0.25 mg/L of these AMPs, respectively; 11 is the negative control; 12 is a positive control, the 20 mg/L ampicillin.

**Figure 9 microorganisms-09-01858-f009:**
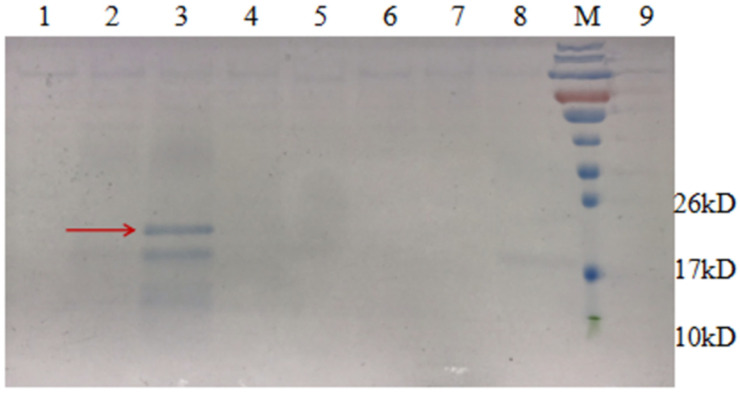
SDS-PAGE (12%) analysis of expression of AMP13 in *Pichia pastoris*. M, protein molecular weight marker. Lane 3, AMP produced by *P. pastoris*.

**Figure 10 microorganisms-09-01858-f010:**
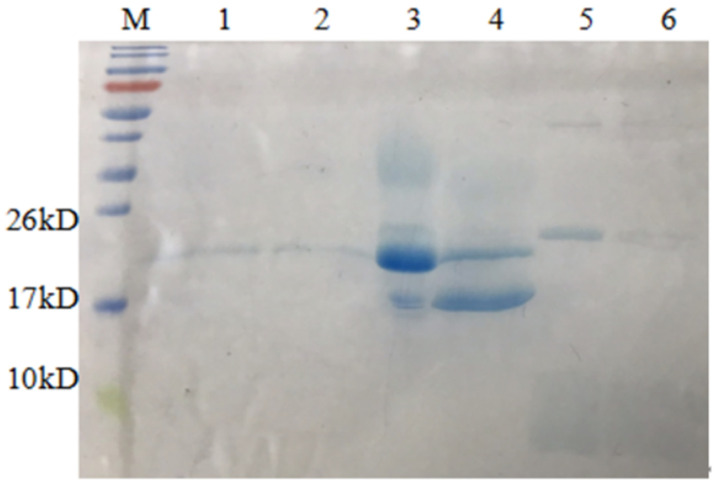
SDS-PAGE (12%) analysis of HRV 3C protease cleavage of AMP13. M, protein molecular weight marker. Lane 1, the supernatant from *P. pastoris*; lane 2, the supernatant from *P. pastoris* heated to 80 °C for 30 min; lane 3, the nickel column before digestion; lane 4, the nickel column after digestion and elution; lane 5, the digestion supernatant; lane 6, HRV 3C Cleavage wash buffer.

**Figure 11 microorganisms-09-01858-f011:**
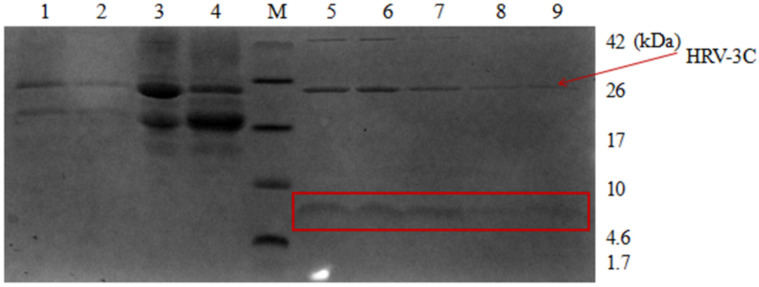
Tricine SDS-PAGE analysis of HRV 3C protease cleavage AMP13. M, protein molecular weight marker. Lane 1, the supernatant from *P. pastoris;* lane 2, the supernatant from *P. pastoris* heated to 80 °C for 30 min; lane 3, the nickel column before digestion; lane 4, the nickel column after digestion and elution; lane 5, the digestion supernatant; lane 6–9, HRV 3C cleavage wash buffer with NaCl.

**Figure 12 microorganisms-09-01858-f012:**
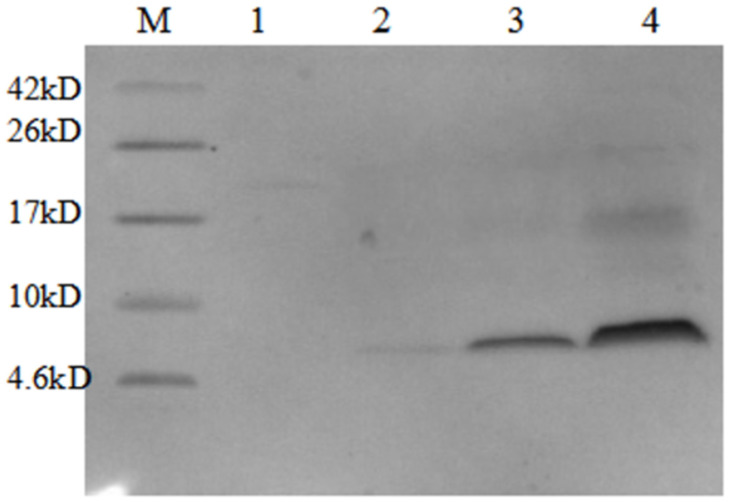
Tricine SDS-PAGE analysis of purification of AMP 13. M, protein molecular weight marker. Lane 1–4, the purified AMP13 produced by *P. pastoris*.

**Figure 13 microorganisms-09-01858-f013:**
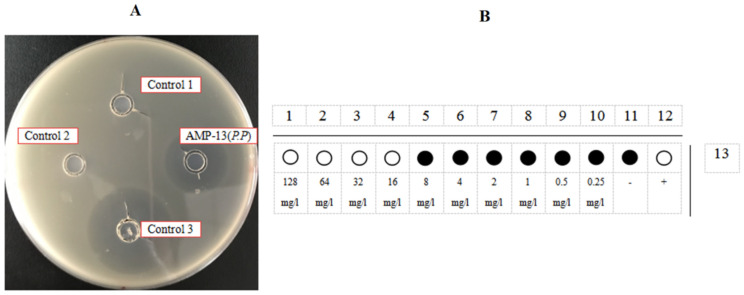
The activity test of AMP13 produced by *P. pastoris*. (**A**). Inhibition zone tests of AMP13. Control 1, HRV 3C Cleavage Buffer; Control 2, HRV 3C Cleavage Buffer with 1 M NaCl; Control 3, Positive control, ampicillin antibiotic at a concentration of 200 mg/L. Every oxford cup contains 100 μL liquid. (**B**). The schematic diagram of a 96-well plate measuring MIC (data taken from three parallel experiments). Horizontal direction is AMP 13 test rows; Vertical direction 1–10 is the final concentration of 128 mg/L to 0.25 mg/L of these AMPs, respectively; 11 is the negative control; 12 is a positive control, the 20 mg/L ampicillin.

**Table 1 microorganisms-09-01858-t001:** Sequences of AMPs.

Serial Number	Amino Acid Sequence (MW: kDa)	Nucleotide Sequence
AP00027(AMP1)	ITPATPFTPAIITEITAAVIA(2.11)	ATTACGCCAGCAACACCCTTCACCCCCGCAATCATCACAGAGATCACGGCGGCGGTCATAGCA
AP00150(AMP2)	ILPWKWPWWPWRR(1.91)	ATTTTACCTTGGAAGTGGCCATGGTGGCCGTGGCGTAGA
AP00155(AMP3)	RGLRRLGRKIAHGVKKYGPTVLRIIRIAG(3.26)	CGGGGCCTTCGTCGTCTGGGCCGTAAGATAGCGCACGGAGTAAAGAAATATGGACCCACCGTACTTAGAATAATTCGTATAGCCGGA
AP00166(AMP4)	GWGSFFKKAAHVGKHVGKAALTHYL(2.71)	GGGTGGGGGAGTTTCTTCAAAAAGGCGGCTCATGTCGGCAAACACGTAGGTAAGGCTGCTCTGACGCATTACTTG
AP00176(AMP5)	ACYCRIPACIAGERRYGTCIYQGRLWAFCC(3.45)	GCTTGTTATTGCAGAATCCCCGCCTGCATTGCTGGAGAGCGTCGCTACGGGACCTGTATATATCAAGGCCGTTTGTGGGCCTTTTGTTGC
AP00276(AMP6)	VFQFLGKIIHHVGNFVHGFSHVF(2.67)	GTATTCCAATTTCTGGGAAAAATAATCCATCATGTAGGGAACTTCGTGCATGGATTTAGCCATGTTTTT
AP00281(AMP7)	GLLRKGGEKIGEKLKKIGQKIKNFFQKLVPQPEQ(3.88)	GGTTTGTTGCGGAAGGGAGGAGAGAAGATAGGTGAAAAACTGAAGAAAATAGGCCAGAAGATTAAGAACTTCTTTCAAAAGTTAGTACCGCAACCTGAGCAA
AP00283(AMP8)	GIINTLQKYYCRVRGGRCAVLSCLPKEEQIGKCSTRGRKCCRRKK(5.16)	GGAATCATCAACACATTACAAAAATACTACTGCCGCGTCAGAGGGGGTCGTTGTGCTGTTTTGAGTTGTCTGCCCAAGGAGGAACAGATAGGAAAATGTTCAACTAGAGGGCGTAAATGCTGTCGCAGAAAGAAA
AP00285(AMP9)	GLLCYCRKGHCKRGERVRGTCGIRFLYCCPRR(3.76)	GGACTGCTGTGTTATTGCCGGAAAGGTCACTGCAAGCGTGGCGAGAGAGTACGGGGAACCTGTGGGATTCGCTTTTTATACTGTTGTCCCAGACGC
AP00351(AMP10)	GLFDVIKKVASVIGGL(1.62)	GGCCTGTTCGACGTGATTAAGAAGGTTGCTTCTGTGATCGGGGGACTT
AP00366(AMP11)	GRFKRFRKKFKKLFKKLSPVIPLLHLG(3.28)	GGGCGGTTCAAACGGTTTAGAAAGAAATTCAAAAAATTATTCAAGAAACTGAGCCCCGTTATTCCGTTGCTTCATCTGGGA
AP00367(AMP12)	GGLRSLGRKILRAWKKYGPIIVPIIRIG(3.13)	GGCGGATTAAGATCGCTGGGCCGGAAAATCCTTCGTGCCTGGAAGAAGTATGGCCCCATAATAGTACCTATCATACGGATTGGG
AP00445(AMP13)	GFCRCLCRRGVCRCICTR(2.11)	GGCTTTTGCAGATGTTTGTGTCGCAGAGGTGTCTGCCGTTGTATTTGCACAAGA
AP00473(AMP14)	FFHHIFRGIVHVGKTIHRLVTG(2.57)	TTTTTTCATCATATTTTCAGAGGCATCGTTCATGTGGGCAAGACTATCCATCGGTTGGTTACAGGC
AP00498(AMP15)	GLVRKGGEKFGEKLRKIGQKIKEFFQKLALEIEQ(3.95)	GGTCTTGTGCGCAAAGGAGGTGAGAAGTTTGGCGAAAAGTTACGCAAAATCGGGCAAAAAATCAAGGAGTTTTTTCAAAAATTAGCTCTGGAGATTGAACAA
AP00505(AMP16)	DSHAKRHHGYKRKFHEKHHSHRGY(3.04)	GACTCCCATGCTAAGAGACACCACGGCTATAAACGTAAGTTTCACGAGAAGCACCATTCTCACCGTGGATAT
AP00513(AMP17)	FLGGLIKIVPAMICAVTKKC(2.11)	TTTCTGGGGGGCCTTATCAAGATCGTACCAGCCATGATATGCGCCGTCACGAAGAAATGT
AP00516(AMP18)	IWLTALKFLGKHAAKHLAKQQLSKL(2.85)	ATATGGTTGACGGCTCTTAAATTTCTGGGAAAACACGCTGCGAAGCACCTTGCGAAACAGCAGCTTAGCAAACTT
AP00519(AMP19)	QWGRRCCGWGPGRRYCRRWC(2.54)	CAATGGGGAAGAAGATGCTGCGGCTGGGGTCCTGGACGCAGATACTGCCGCCGTTGGTGT
AP00538(AMP20)	WLNALLHHGLNCAKGVLA(1.93)	TGGCTGAATGCCTTATTACACCACGGTCTGAACTGTGCCAAAGGTGTACTGGCC

## Data Availability

All relevant data of this study are presented. Additional data can be provided upon request.

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
