# Peer review of "A Method for Rapid Screening, Expression, and Purification of Antimicrobial Peptides"

_microorganisms, 2021, doi:10.3390/microorganisms9091858_

Round 1

Reviewer 1 Report

Dear Authors, 

A very interesting manuscript. The methods and results are well described, but I have questions and suggestions like this:

  1. How was the concentration of the tested peptides determined?
  2. Mapping the expression plasmids would be useful
  3. It would be worth supplementing the description on the methodology of electrophoresis, acrylamide concentration, agarose, etc.
  4. The names of microorganisms should be written in italics
  5. General note: in my opinion the results are described too extensively, maybe some of them can be included as Supplementary Materials?

Reviewer 2 Report

The submitted manuscript “A method for rapid screening, expression, and purification of antimicrobial peptides” by Zhang et al described a method for the rapid screening, expression and purification of antimicrobial peptides (AMPs), followed by the activity determination. At this stage, I recommend this manuscript be published after minor revision.

  1. Some repeating expressions or abbreviations, for example, antimicrobial peptides (AMPs) was frequently used in the text but not consistent.
  2. Page 1 line 31, they described AMPs in general. The recent multiple excellent reviews on AMPs should be included (Lancet Infect Dis 2020; 20: e216–30, https://doi.org/10.1016/S1473-3099(20)30327-3, Chem. Soc. Rev., 2021,50, 4932-4973 https://doi.org/10.1039/D0CS01026J).
  3. Page 1 line 43, they summarised the advantages of AMPs. They should also mention the limitations of AMPs in comparison with conventional antibiotics.
  4. Page 4 table 1, they should add the AMP numbers and theoretical molecular weight into the table to corresponding AMP 1, 2, 3, ….
  5. Page 10 figure 5, it is better to have the mass spec to characterise the purified AMPs.
  6. Page 11 line 278, they claimed “AMP13 and AMP16 was 24.4 uM”. However, it is not correlated with figure 7.
